# Hierarchical Multi-label Learning for Incremental Multilingual Text Recognition

## ABSTRACT

Multilingual text recognition (MLTR) is increasingly essential for facilitating cultural communication. However, existing methods often struggle with retaining previous knowledge when learning new languages. A straightforward solution is performing incremental learning (IL) on MLTR tasks. However, it ignores the shared words and characters across incremental languages, which we first term as an *incremental sharing* problem. Motivated by this observation, we propose a **H**ier**A**rchical **M**ulti-label learning framework for **M**ultilingual t**E**xt **R**ecognition, termed HAMMER. An online knowledge analysis is designed to identify shared knowledge and provide corresponding multi-label language supervision. Specifically, only words and characters appearing simultaneously in multiple languages are considered shared knowledge. Additionally, to further capture language dependencies, we introduce a hierarchical language evaluation mechanism to predict language scores at word and character levels. These scores, supervised by the knowledge analysis, guide the specific recognizers to effectively utilize both old and new language knowledge, thereby mitigating catastrophic forgetting caused by imbalanced rehearsal sets. Extensive experiments conducted on benchmark datasets, MLT17 and MLT19, show that HAMMER exhibits remarkable results and outperforms other state-of-the-art approaches.

## CCS CONCEPTS

• **Computing methodologies** → *Computer vision tasks*.

## KEYWORDS

Multilingual text recognition; Incremental learning; Catastropic forgetting; Incremental sharing.

## 1 INTRODUCTION

Scene text recognition (STR) [10, 34, 46, 51] involves reading character sequences from scene text images and is a challenging task in computer vision. Recent advancements in deep learning have facilitated the progress of text recognition technology, making it widely applicable in various real-world scenarios such as autonomous driving, machine translation, and image-text retrieval. However, most existing methods primarily focus on Latin script [4, 43, 45] and struggle to handle multilingual scenarios effectively. As cross-cultural communications become more prevalent, there is a growing

**Unpublished working draft. Not for distribution.**

need for multilingual text recognition (MLTR) [13, 14, 42]. A robust MLTR model should be capable of recognizing multiple languages simultaneously and continuously learning new languages. Existing MLTR methods often train a multilingual recognizer by mixing samples from all languages together [5, 24]. When encountering a new language, these methods incorporate the new samples into the original old data to retrain the recognizer, leading to increased computational complexity [27, 41] and potential storage constraints.

Incremental learning (IL) [11, 18, 22, 49] is a powerful paradigm for continuously learning new data while retaining knowledge of old data. It is promising to yield fine recognizers when incremental methods are applied to MLTR tasks. However, like some incremental learning dilemmas, incremental multilingual recognizers also face the issue of catastrophic forgetting, where the recognizer tends to forget the old languages while learning new ones.

In incremental methods, replay-based ones [6, 28, 37] have shown promising performance by obtaining a small amount of old data, referred to as the rehearsal set. Conventional replay-based methods typically sample old classes uniformly to create a fixed-capacity rehearsal set. Nevertheless, this operation is not directly applicable to MLTR tasks. In MLTR, languages are treated as incremental tasks, and the characters composing each word are considered incremental classes. Unlike other tasks where classes are sampling units (e.g., image classification), in MLTR tasks, the sampling unit is the word itself, not the character classes. Consequently, there is no guarantee that the rehearsal set covers all character classes of old languages, resulting in a rehearsal-imbalance problem [52]. This imbalanced rehearsal set would exacerbate catastrophic forgetting, significantly impacting recognition performance in old languages.

Existing replay-based incremental MLTR methods [38, 52] maintain awareness of old knowledge by loading corresponding specific parameters after determining the belonging language class of the entire words. An inspired work, MRN [52], introduces a multiplexed routing network that predicts language scores of entire words to weight the various specific recognizers for decoding character sequences. Although these methods alleviate catastrophic forgetting caused by the imbalanced rehearsal set to some extent, they still suffer from two limitations. (1) Fine-grained information at the character level is ignored. A word is composed of multiple characters, and in addition to the whole word, individual characters also contain valuable information. (2) Shared knowledge among incremental languages is neglected. In MLTR tasks, defining a sample as belonging strictly to a specific language is inaccurate since a sample may appear in multiple languages simultaneously. For instance, words composed solely of digits belong to both English and Chinese. Simply assuming these words belong exclusively to either English or Chinese would lead to knowledge misalignment. In this paper, we innovatively refer to this phenomenon as the ***incremental sharing*** problem.

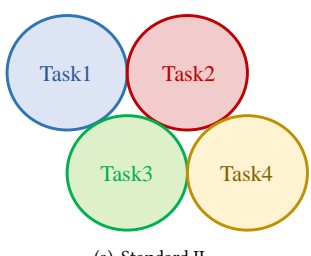 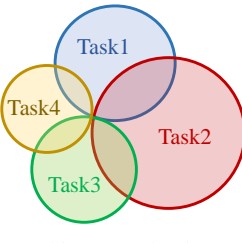

(a) Standard IL      (b) MLTR IL (ours)

**Figure 1: Diagram of incremental settings. Colors indicate incremental tasks. The circle size indicates the number of incremental classes or instances. The overlap in the MLTR IL setting indicates character sharing or word sharing.**

Different from the standard IL setting shown in Fig. 1(a), there is an overlap in the MLTR IL setting depicted in Fig. 1(b). Overlap indicates the intersection between the content of incremental tasks, i.e., incremental sharing, which is mainly manifested in two aspects: 1) Incremental character classes sharing, where a character class exists in multiple languages, and 2) Incremental word instances sharing, where a word instance is present in more than one language. Particularly, we define words as shared only when each character of the word appears in multiple identical languages simultaneously. To demonstrate the incremental sharing problem, we further conduct a statistical analysis in two popular multilingual text datasets, MLT17 [25] and MLT19 [24]. As indicated in Table 1, 20%-55% of incremental character classes are shared, and 0.3%-26% of incremental word instances are shared. Therefore, exploring the common knowledge between these shared characters and words is crucial for incremental MLTR tasks.

Motivated by the above observation, we propose a **H**ier**A**rchical **M**ulti-label learning framework for incremental **M**ultilingual t**E**xt **R**ecognition, named HAMMER. It comprises two stages: specific language learning and multilingual learning. Initially, the former stage involves training a specific text recognizer for each incremental language, similar to standard STR models. Subsequently, in the multilingual learning stage, an online knowledge analysis module is devised to determine whether samples belong to shared or new knowledge, providing corresponding multi-labeled and single-labeled knowledge supervision. Additionally, a hierarchical language estimation mechanism is proposed to predict language scores at both the word and character levels using a DomainMLP that processes character features extracted by the frozen specific recognizers. These scores, supervised by the knowledge analysis module, guide both old and new recognizers in decoding character sequences. By leveraging multi-label learning of shared knowledge at the word and character levels, our HAMMER framework can adequately exploit the old and new knowledge, thereby alleviating catastrophic forgetting stemming from imbalanced rehearsal sets and enhancing recognition performance.

Our contributions are summarized as follows:

- We first identify the incremental sharing problem specific to incremental MLTR tasks, which differs from conventional incremental learning settings.

**Table 1: Statistics of characters and words on MLT17 and MLT19 datasets. *Total* represents the number of identical languages in both datasets.**

| Statistics | | Task1 Chinese | Task2 Latin | Task3 Japanese | Task4 Korean | Task5 Arabic | Task6 Bangla |
|---|---|---|---|---|---|---|---|
| Chars | Total | 2086 | 354 | 1733 | 1186 | 74 | 113 |
| | Shared | 0 | 81 | 942 | 274 | 15 | 51 |
| | Rate | 0 | 0.229 | 0.544 | 0.231 | 0.203 | 0.451 |
| Words (train) | Total | 5584 | 100332 | 9933 | 11738 | 7941 | 6779 |
| | Shared | 0 | 13144 | 2517 | 1381 | 20 | 381 |
| | Rate | 0 | 0.131 | 0.253 | 0.118 | 0.003 | 0.056 |
| Words (test) | Total | 851 | 16955 | 1940 | 1909 | 1453 | 1106 |
| | Shared | 0 | 2395 | 500 | 237 | 4 | 53 |
| | Rate | 0 | 0.141 | 0.258 | 0.124 | 0.003 | 0.048 |

- An effective incremental MLTR framework, HAMMER, is proposed, which is able to leverage old knowledge adequately and mitigate catastrophic forgetting by identifying and learning shared knowledge at both word and character levels.
- Extensive experiments are conducted on two mainstream MLTR datasets. The proposed framework gains significant results and establishes new state-of-the-art (SOTA) ones.

## 2 RELATED WORK

### 2.1 Deep Learning-based STR

Mainstream deep learning-based STR methods typically employ three types of decoders: CTC [9, 12, 30], RNN [3, 4, 33], and transformer [8, 23, 26, 35, 40] decoders. TRBA [3], for instance, is a representative STR baseline with TPS [31] transformation. ABINet [8] is a transformer-based baseline explicitly exploring language context. Recently, with the advances in large language models (LLM), CLIP-based methods [1, 36, 50] have shown promising results on various text-related tasks.

Multilingual text recognition [13, 24, 25, 42], a sub-field of STR, aims to recognize multiple languages simultaneously and continuously learn new languages. Early solutions [52] involve training models on all language data together, leading to biases towards samples-rich languages and poor performance for samples-scarcity languages. Alternatively, some methods [14, 38] introduce an auxiliary language prediction network to determine the language class, and then specific recognizers or private parameters are loaded to predict character sequences. While similar to our pipeline, these methods do not address the incremental sharing problem, an essential property of MLTR tasks.

### 2.2 Incremental Learning

Incremental learning [21, 22, 32, 49] is crucial for continuously processing new data while retaining previous knowledge, thus overcoming catastrophic forgetting. Mainstream incremental learning methods fall into three categories. **Regularization-based methods** prevent the overwriting of old knowledge by imposing constraints on the loss of new tasks. For instance, LwF [18] ensures similar predictions between old and new models on new tasks

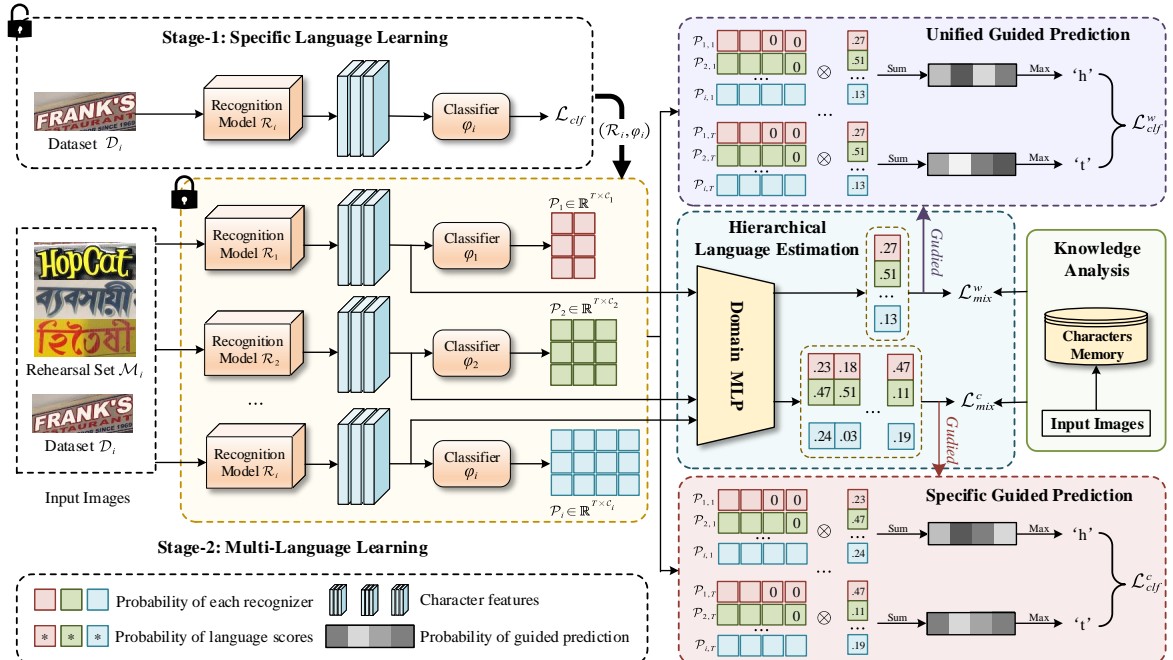

**Figure 2: The pipeline of our HAMMER. Stage 1 involves training specific language recognizers. During incremental multilingual learning for task $i$, we perform online knowledge analysis at word and character levels to determine whether samples are shared and provide corresponding multi-label supervision. Then, the hierarchical language estimation predicts the belonging language scores at word and character levels. These scores guide specific recognizers in predicting character sequences.**

through knowledge distillation. An improved algorithm, EWC [15], introduces a generalized parameter constraint method based on the Bayesian framework, adding an additional parameter regular loss. **Replay-based methods** retain a representative portion of old data to review previously learned knowledge. iCaRL [27], for example, employs distillation loss to update model parameters while allowing the use of old data. CLEAR [28] dynamically adjusts the amount of old data retained, avoiding the linear growth of computational cost seen in LwF. **Expansion-based methods** expand a set of task-specific parameters for each incremental task. PGN [29] creates a task-specific network and transfers knowledge among them through horizontal connections. However, as the number of tasks increases, memory usage grows linearly. To mitigate memory overhead, DEN [44] and RCL [39] only expand network width when capacity is insufficient. This paper employs the incremental learning paradigm to perform MLTR tasks.

## 3 OUR METHOD

### 3.1 Problem Definition

Given incremental languages $\mathcal{D} = \{\mathcal{D}_i\}_{i=1}^{I}$, we aim to improve performance on the current language $\mathcal{D}_i$ and maintain perception of old languages $\tilde{\mathcal{D}}_{i-1} = \{\mathcal{D}_k\}_{k=1}^{i-1}$, where $\mathcal{D}_i = \{(x_{i,j}, y_{i,j})\}_{j=1}^{N_i}$ is the dataset of task $i$, with $N_i$ being the number of training samples. Each $y_{i,j} = \{y_{i,j,1}, y_{i,j,2}, \cdots, y_{i,j,T}\} \in \mathcal{D}_i$ is a sequence label within the label set $C_i$, where $T$ denotes the pre-defined maximum decoding length. The incremental sharing problem identified innovatively by

us differs from conventional incremental learning settings in two ways: (1) incremental character classes sharing, i.e., $C_i \cap \tilde{C}_{i-1} \neq \emptyset$, where $\tilde{C}_{i-1} = \cup_{k=1}^{i-1} C_k$ denote the label space of all character classes up to task $i$-1, and (2) incremental word instances sharing, i.e., $\mathcal{D}_i \cap \tilde{\mathcal{D}}_{i-1} \neq \emptyset$. Due to the presence of shared characters and words, the replay-based incremental paradigm is naturally chosen for its access to some of the old data, referred to as the rehearsal set $\mathcal{M}_i = sampler(\tilde{\mathcal{D}}_{i-1})$, thus facilitating the exploration of dependency relationships between the old and the new languages.

From the perspective of language classes, these shared characters and words are multi-labeled due to their presence in multiple languages. Therefore, multi-label learning is utilized to optimize the belonging language classes of shared samples. Furthermore, each character composing a word contains valuable information besides the entire word. Consequently, hierarchical analysis at the word and character levels benefits recognition performance. Based on the motivations above, we propose a two-stage hierarchical multi-label learning framework, HAMMER. An overview of the HAMMER is illustrated in Fig. 2.

### 3.2 Specifc Language Learning

In stage 1, a specific recognizer is trained using the language dataset $\mathcal{D}_i$ corresponding to the task $i$. Specifically, we denote the feature extractor and classifier as $\mathcal{R}_i$ and $\varphi_i$, respectively, and $\mathcal{P}_i$ as the predicted probability distribution. Given the $(x, y) \in \mathcal{D}_i$, we obtain $\mathcal{P}_i(x) = \varphi_i(\mathcal{R}_i(x))$. The standard cross-entropy loss $\mathcal{L}_{clf}$ is

employed to optimize the specific recognizer,

$$\mathcal{L}_{clf} = \mathbb{E}_{(x,y)\sim\mathcal{D}_i}\left[\sum_{t=1}^{T} -y\log\mathcal{P}(y_t|x)\right], \tag{1}$$

where $\mathcal{P}(y_t|x)$ represents the predicted probability of the output being $y_t$ at time step $t$.

## 3.3 Knowledge Analysis

In MLTR tasks, shared characters and words appear in multiple incremental languages. As mentioned previously, we term this phenomenon in MLTR tasks as *incremental sharing* problem. Thoroughly exploring the relationship between shared characters and words is crucial for enhancing MLTR performance. Due to the lack of comprehensive belonging language supervision of the shared characters and words in the original datasets, we first design an online knowledge analysis to distinguish between new and shared knowledge. New knowledge refers to characters or words unique to the current task, whereas shared ones encompass those that also appear in other task(s). Precisely, as illustrated in Fig. 3(a), we classify a word instance as shared only where each constituent character is also present simultaneously in other identical task(s).

Formally, let $(x, y) \in \mathcal{D}_i \cup \mathcal{M}_i$ denote an image-text pair to be recognized in incremental task $i$. Initially, the character memory is updated dynamically by adding current character classes $C_i$ to the memory. Then, the character $y_t$ in the sequence $y = \{y_1, y_2, \cdots, y_T\}$ can be encoded based on its associated language classes. If its encoding length is 1, the character is new, existing solely in the current task. Conversely, the character is deemed shared if the encoding length is greater than 1, indicating multiple languages. The knowledge type of character $y_t$ is formulated as,

$$\mathcal{K}(y_t|x) = \begin{cases} New, & len(\mathcal{E}(y_t)) = 1 \\ Shared, & len(\mathcal{E}(y_t)) > 1 \end{cases}, \tag{2}$$

where $\mathcal{E}(\cdot)$ denotes the language encoding, and $len(\cdot)$ is a calculating length fucntion. A word encoding is obtained by intersecting the encodings of its constituent characters. Similarly, this word is new if its encoding length is 1 and shared if it exceeds 1. The knowledge type of word $y$ is formulated as,

$$\mathcal{K}(y|x) = \begin{cases} New, & len(\mathcal{E}(y)) = 1 \\ Shared, & len(\mathcal{E}(y)) > 1 \end{cases}, \tag{3}$$

where $\mathcal{E}(y) = \mathcal{J}(\mathcal{E}(y_1), \cdots, \mathcal{E}(y_T))$. The $\mathcal{J}(\cdot)$ is the intersection of all character encodings. As depicted in Fig. 3(b), if any new character is present in a word, the word encoding after character intersection must be of length 1, signifying that the word is new. In essence, a character unique to the current language implies that the word is also new, which is consistent with our design.

Based on the determined knowledge types above, this module plays a crucial role in providing corresponding knowledge supervision. For new knowledge, the supervision labels are one-hot encodings. Multi-label optimization aligns precisely with the purpose of shared knowledge learning. Thus, multi-label learning is employed for shared knowledge optimization. It is worth noting that multiple labels require transformation based on the principles of multi-label learning. The language class labels of character $y_i$ and word $y$ are

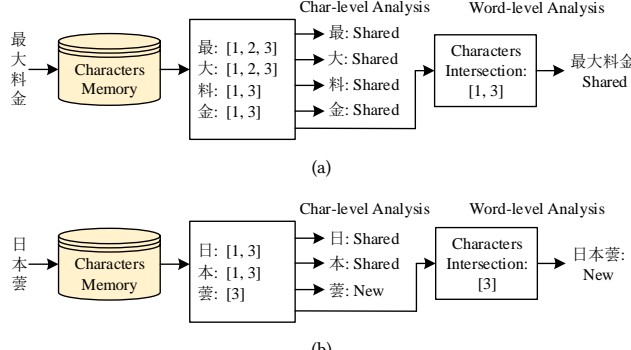

Figure 3: Illustration of knowledge analysis. (a): each character belongs to shared knowledge, and then the word belongs to shared knowledge. (b): as long as one character belongs to new knowledge, the word belongs to new knowledge.

defined as,

$$\begin{cases} \mathcal{N}^c(y_t|x) = OneHot(\mathcal{E}(y_t)), & \mathcal{K}(y_t|x) = New \\ \mathcal{H}^c(y_t|x) = TransMulti(\mathcal{E}(y_t)), & \mathcal{K}(y_t|x) = Shared \end{cases}, \tag{4}$$

$$\begin{cases} \mathcal{N}^w(y|x) = OneHot(\mathcal{E}(y)), & \mathcal{K}(y|x) = New \\ \mathcal{H}^w(y|x) = TransMulti(\mathcal{E}(y)), & \mathcal{K}(y|x) = Shared \end{cases}, \tag{5}$$

where $\mathcal{N}^*$ and $\mathcal{H}^*$ represent the new and shared knowledge label sets, respectively. The $OneHot(\cdot)$ is the one-hot encoding, and $TransMulti(\cdot)$ denotes the multi-label transformation.

## 3.4 Hierarchical Language Estimation

Replay-based incremental methods alleviate catastrophic forgetting by preserving a rehearsal set. Typically, this rehearsal set uniformly selects samples from each incremental class to ensure adequate coverage of all old classes. However, achieving even selection across all character classes is challenging in incremental MLTR tasks, where word instances are the sample objects. This selection way may lead to some old character classes being excluded from the rehearsal set. This imbalanced rehearsal set may hinder the robust recognition. Compared to a large number of character classes, language classes are relatively tiny. Therefore, training a language class predictor might be feasible for enhancing the performance.

Existing incremental MLTR methods design a language predictor solely at the word level. We argue that this word-level prediction is inadequate for sequential tasks, particularly text recognition. Firstly, considering the shared words between multiple languages, it is premature to assume that a word solely belongs to a specific language, as discussed in Section 3.3. Secondly, judging the language only at the word level disregards the abundant information at the fine-grained character level. For instance, as shown in Fig. 3(b), if we simply assume the entire word is Japanese (encoding [3]), the Japanese recognizer will be favored, disregarding that the first and second characters belong to both Chinese and Japanese (encoding [1, 3]), which results in under-utilization of the feature extraction capability of the Chinese recognizer. Therefore, it is essential to analyze the language classes of each character in addition to those of the entire word.

 

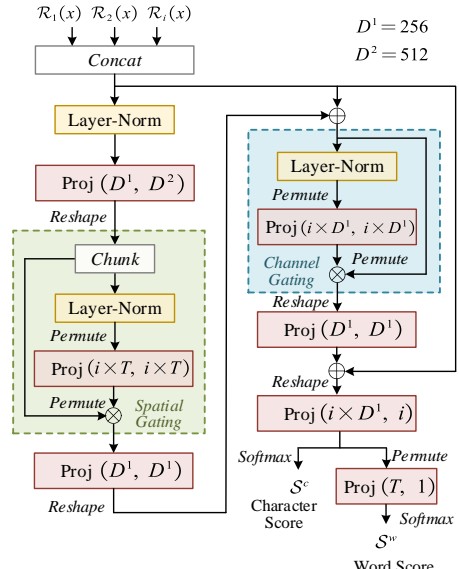

**Figure 4: Detailed architecture of DomainMLP. The Spatial Gating and Channel Gating explore the relationship between characters extracted by multiple specific recognizers.**

To achieve this, we introduce a hierarchical language estimation mechanism to analyze the belonging languages of words and characters, respectively. Technically, the features $\{\mathcal{R}_1(x), \cdots, \mathcal{R}_i(x)\}$ extracted by the frozen specific recognizers are simultaneously fed into a DomainMLP. The architecture of the DomainMLP is detailed in Fig. 4. By capturing the latent correlation between features in both spatial and channel dimensions, the DomainMLP predicts the word score $\mathcal{S}^w(x) \in \mathbb{R}^{1 \times i}$ and the character score $\mathcal{S}^c(x) \in \mathbb{R}^{T \times i}$, where $T$ is the maximum decoding length. These scores are optimized by cross-entropy loss $\mathcal{L}_{CE}$ for new knowledge and multi-label loss $\mathcal{L}_{mul}$ for shared knowledge. Thus, we define the mixed loss $\mathcal{L}_{mix}$ as,

$$\mathcal{L}^*_{mix} = \mathcal{L}^*_{CE} + \mathcal{L}^*_{mul}, \tag{6}$$

$$\mathcal{L}^*_{CE} = \mathbb{E}_{x \sim \mathcal{D}_i \cup \mathcal{M}_i}\left[-\mathcal{N}^*(x) \log \mathcal{S}^*(x)\right], \tag{7}$$

$$\mathcal{L}^*_{mul} = \mathbb{E}_{x \sim \mathcal{D}_i \cup \mathcal{M}_i}\left[\sum_{m,n} \frac{\max(0, 1 - (\mathcal{S}^*[\mathcal{H}^*[m]] - \mathcal{S}^*[n]))}{\mathcal{S}^*(x).size(-1)}\right], \tag{8}$$

where $* = \{w, c\}$, denoting word and character levels. $\mathcal{N}^*$ and $\mathcal{H}^*$ represent language class labels for the new and shared knowledge obtained from Eq. 4 and Eq. 5. In addition to being supervised by the knowledge analysis module, these language scores also serve as guidance for the specific recognizers trained in stage 1.

## 3.5 Unified and Specific Guided Predictions

The initial probabilities of specific recognizers can be considered generalizations to other languages. Typically, a specific recognizer performs better on words or characters seen in their own languages but worse on unseen ones. Given the shared words and characters, there may be multiple specific recognizers with better generalization rather than just one. Fully leveraging the feature extraction

capabilities of these recognizers and mining the dependencies between languages to collaborate on recognition are crucial. Consequently, based on the language scores from the DomainMLP, we propose unified guided prediction for word level and specific guided prediction for character level. The *unified* indicates that all characters within a word have the same weight to recognizers, while the *specific* indicates that the attention weight varies among characters.

More formally, given $(x, y) \in \mathcal{D}_i \cup \mathcal{M}_i$, we obtain a set of initial probabilities $\{\mathcal{P}_1(x) \in \mathbb{R}^{T \times C_1}, \cdots, \mathcal{P}_i(x) \in \mathbb{R}^{T \times C_i}\}$. These probabilities are padded with zeros to align with the dimensions $\tilde{C}_i$ of the union of all classes, denoted as $\tilde{C}_i = \cup_{k=1}^i C_k$. Thus, the aligned probabilities become $\{\mathcal{P}_1(x) \in \mathbb{R}^{T \times \tilde{C}_i}, \cdots, \mathcal{P}_i(x) \in \mathbb{R}^{T \times \tilde{C}_i}\}$. Then, the word-level probability $\tilde{\mathcal{P}}^w(x)$ after being guided by the word score $\mathcal{S}^w(x) \in \mathbb{R}^{1 \times i}$ in unified guided prediction is,

$$\tilde{\mathcal{P}}^w(x) = \sum_{k=1}^i \left(\mathcal{P}_k(x) \times \mathcal{S}_k^w(x)\right). \tag{9}$$

Similarly, the guided character-level probability $\tilde{\mathcal{P}}^c(x)$ by the character score $\mathcal{S}^c(x) \in \mathbb{R}^{T \times i}$ in specific guided prediction is,

$$\tilde{\mathcal{P}}^{c,(t)}(x) = \sum_{k=1}^i \left(\mathcal{P}_k^{(t)}(x) \times \mathcal{S}_k^{c,(t)}(x)\right), \tag{10}$$

where superscript $(t)$ indicates the operation is performed at each time step $t$. In essence, both Eq. 9 and Eq. 10 employ a soft voting way to obtain the final guided probabilities. More discussion about the voting mechanism is provided in Sec. 4.3.2.

These guided probabilities are optimized using standard cross-entropy to enhance recognition performance,

$$\mathcal{L}_{clf}^w = \mathbb{E}_{(x,y) \sim \mathcal{D}_i \cup \mathcal{M}_i}\left[\sum_{t=1}^T -y \log \tilde{\mathcal{P}}^w(y_t|x)\right], \tag{11}$$

$$\mathcal{L}_{clf}^c = \mathbb{E}_{(x,y) \sim \mathcal{D}_i \cup \mathcal{M}_i}\left[\sum_{t=1}^T -y \log \tilde{\mathcal{P}}^c(y_t|x)\right], \tag{12}$$

where $\tilde{\mathcal{P}}^w(y_t|x)$ and $\tilde{\mathcal{P}}^c(y_t|x)$ represent the word and character probabilities of the output being $y_t$ at time step $t$.

## 3.6 Total Training Loss

As shown in Eq. 1, only $\mathcal{L}_{clf}$ is used to optimize the specific language learning in stage 1. In stage 2, the loss comprises two components: the classification loss, $\mathcal{L}_{clf}^w$ and $\mathcal{L}_{clf}^c$, and the language prediction loss, $\mathcal{L}_{mix}^w$ and $\mathcal{L}_{mix}^c$. Therefore, the total training loss in the multilingual learning stage is defined as,

$$\mathcal{L} = \mathcal{L}_{clf}^w + \mathcal{L}_{clf}^c + \lambda\left(\mathcal{L}_{mix}^w + \mathcal{L}_{mix}^c\right), \tag{13}$$

where $\lambda$ is a trade-off parameter.

To better illustrate the training process of stage 2, the pseudocode is summarized in Algorithm 1.

# 4 EXPERIMENTS AND RESULTS

## 4.1 Datasets and Implementation Details

**Datasets:** Current mainstream multilingual text recognition datasets are MLT17 [25] and MLT19 [24]. **MLT17** is a natural scene dataset with blur, occlusion, and distortion challenges. It comprises 6 scripts:

**Algorithm 1:** Multi-Language Learning in Stage 2

**Input:** incremental task ID $i$, dataset $\mathcal{D}_i$, label set $C_i$, the specifc recognizers $\{\mathcal{R}_k, \varphi_k\}_{k=1}^{i}$ trained in stage 1, the size of rehearsal set $M$.

**Output:** *DomainMLP* parameterized by $\theta_{mlp}$.

1  // Construct rehearsal set

2  **for** $k \leftarrow 1$ *to* $i-2$ **do**

3     | // *Sampler*$(a, b)$: randomly select $b$ samples from set $a$

4     | $\mathcal{D}_k^{\mathcal{M}_i} = Sampler\,(\mathcal{D}_k^{\mathcal{M}_{i-1}}, \frac{M}{i-1})$

5     | $\mathcal{M}_i \leftarrow Add\,(\mathcal{D}_k^{\mathcal{M}_i})$

6  **end**

7  $\mathcal{D}_{i-1}^{\mathcal{M}_i} = Sampler\,(\mathcal{D}_{i-1}, \frac{M}{i-1})$

8  $\mathcal{M}_i \leftarrow Add\,(\mathcal{D}_{i-1}^{\mathcal{M}_i})$

9  **while** *not at the end of training* **do**

10    | // Knowledge Analysis

11    | Update Character Memory: $Add\,(C_i)$

12    | $\mathcal{N}^c, \mathcal{H}^c = CharAnalysis\,(\mathcal{D}_i \cup \mathcal{M}_i)$

13    | $\mathcal{N}^w, \mathcal{H}^w = WordAnalysis\,(\mathcal{D}_i \cup \mathcal{M}_i)$

14    | // Hierarchical Language Estimation

15    | $\mathcal{S}^w, \mathcal{S}^c = DomainMLP\,(\mathcal{D}_i \cup \mathcal{M}_i)$

16    | Compute $\mathcal{L}_{mix}^w\,(\mathcal{S}^w, \mathcal{N}^w, \mathcal{H}^w)$ by Eq. 6

17    | Compute $\mathcal{L}_{mix}^c\,(\mathcal{S}^c, \mathcal{N}^c, \mathcal{H}^c)$ by Eq. 6

18    | // Unified and Specifc Guided Prediction

19    | Get $\tilde{\mathcal{P}}^w$ by Eq. 9

20    | Get $\tilde{\mathcal{P}}^c$ by Eq. 10

21    | Compute $\mathcal{L}_{clf}^w$ by Eq. 11

22    | Compute $\mathcal{L}_{clf}^c$ by Eq. 12

23    | // Compute Total Loss

24    | Compute $\mathcal{L} = \mathcal{L}_{clf}^w + \mathcal{L}_{clf}^c + \lambda\,(\mathcal{L}_{mix}^w + \mathcal{L}_{mix}^c)$

25    | UpdateParams $(\mathcal{L}, \theta_{mlp})$

26  **end**

27  Return optimized parameters $\theta_{mlp}$.

---

Chinese, Latin, Japanese, Korean, Arabic, and Bangla, totaling 68,613 training and 16,255 test instances. We divide the scripts into 6 incremental tasks. **MLT19** is another real scene dataset containing 7 scripts totaling 89,177 instances. To maintain consistency with MLT17, we exclude the Hindi script language. Consequently, we divide the remaining 6 scripts into 6 incremental tasks. Following MRN [52], we merge languages belonging to the same scripts in both MLT17 and MLT19 into single incremental tasks. The statistics of the combined samples are presented in Table 1. To mitigate dataset variance, we evenly select half of the training samples from MLT17 and MLT19 to form the mini-batch data.

**Implementation Details:** Three representative STR methods with different decoding ways, CRNN [30], SVTR [7], and TRBA [3], are selected to deploy our framework. The default parameters are kept consistent with those in the original papers. The size of the rehearsal set $M$ is set to 2,000, and the decoding length $T$ is set to 25. Stages 1 and 2 are trained for 20,000 iterations with a batch size of 128. To ensure a fair comparison, we set the default incremental order as Chinese→Latin→Japanese→Korean→Arabic→Bangla. All experiments are conducted using PyTorch on two NVIDIA GeForce RTX 2080Ti GPUs.

## 4.2 Comparison with SOTAs

We conduct a comprehensive comparison between our proposed HAMMER and other incremental methods. Similar to MRN [52], we deploy HAMMER on three STR methods with different decoding ways: CTC-based CRNN [30], RNN-based TRBA [3], and transformer-based SVTR [7]. Specifically, we utilize the output before CTC of CRNN, the output of TRBA after attention, and the output of the penultimate layer of the transformer decoder of SVTR as character features, respectively. Meanwhile, four popular IL methods, i.e., LwF [18], EWC [15], WA[48], and DER [41], are chosen to perform MLTR tasks, where the recognition results are derived from MRN. We establish an *UpperBound* model trained on all samples, representing the performance upper limit of our task. The *Baseline* model is trained solely on the current task without any incremental optimization and is tested directly on both old and new tasks. The results of task $i$ ($i>1$) represent the average performance over both old and new tasks. The results of task 1 solely indicate the recognition ability of the current STR models (CRNN, SVTR, and TRBA) and cannot be used as an indicator to evaluate the incremental method. The *AVG* results represent the average performance across all tasks. The experimental results are summarized in Table 2, demonstrating that:

- The poor performance of the baseline model highlights significant differences in incremental languages. Consequently, models trained solely on new tasks experience degradation when tested on old tasks, leading to low average results.
- The sub-optimal results of CRNN-based HAMMER in task 2 may be attributed to insufficient discrimination in character features between old and new tasks, mainly when the number of incremental tasks is small. However, as the number of tasks increases, HAMMER effectively handles incremental sharing between old and new tasks, improving average performance.
- HAMMERs, deployed across the three STR methods, exhibit substantial improvement compared to other incremental methods. Stage 1 enables the exploration of specific knowledge within each language, while stage 2 delves into incremental sharing between languages. This cooperation facilitates deeper mining of old knowledge, thereby mitigating catastrophic forgetting.

## 4.3 Ablation Study

Due to the effectiveness of TRBA architecture, it is selected to perform ablation experiments. For brevity, only the results for *Task6* and *AVG* are provided in the following experimental analysis unless otherwise specified.

*4.3.1 Effect of Each Component.* Some experiments are conducted to validate the effectiveness of each proposed module. The results are summarized in Table 3. We design several variants of HAMMER, indexed from top to bottom as 1-7. *Word-level* and *Char-level* indicate whether the language evaluation module predicts language scores at the word and character levels, respectively. *New* denotes considering all words and characters as new knowledge, optimized

**Table 2: Comparison of our proposed HAMMER and existing SOTA methods on incremental MLTR tasks.**

|  | Methods | MLT17 | | | | | | | MLT19 | | | | | | |
|---|---|---|---|---|---|---|---|---|---|---|---|---|---|---|---|
|  |  | Task1 Chinese | Task2 Latin | Task3 Japanese | Task4 Korean | Task5 Arabic | Task6 Bangla | AVG | Task1 Chinese | Task2 Latin | Task3 Japanese | Task4 Korean | Task5 Arabic | Task6 Bangla | AVG |
| CRNN [30] | UpperBound | - | - | - | - | - | - | 92.10 | - | - | - | - | - | - | 84.90 |
|  | Baseline [52] | 91.10 | 51.70 | 51.00 | 37.20 | 29.30 | 22.30 | 47.10 | 85.10 | 49.60 | 46.50 | 35.50 | 27.60 | 20.70 | 44.20 |
|  | Baseline(Ours) | 91.12 | 43.52 | 25.71 | 23.84 | 20.87 | 15.89 | 36.83 | 85.09 | 39.24 | 25.07 | 22.31 | 19.95 | 13.91 | 34.26 |
|  | LwF(TPAMI2017) [18] | 91.10 | 53.70 | 53.40 | 38.20 | 29.70 | 23.70 | 48.30 | 85.10 | 51.60 | 49.20 | 36.50 | 27.70 | 22.00 | 45.30 |
|  | EWC(PNAS2017) [15] | 91.10 | 56.50 | 50.40 | 37.20 | 30.50 | 21.50 | 47.90 | 85.10 | 55.50 | 46.30 | 35.80 | 28.80 | 19.90 | 45.20 |
|  | WA(CVPR2020) [48] | 91.10 | 54.60 | 48.70 | 38.20 | 28.50 | 23.10 | 47.40 | 85.10 | 52.20 | 44.30 | 36.70 | 26.80 | 21.60 | 44.40 |
|  | DER(CVPR2021) [41] | 91.10 | 76.30 | 55.80 | 46.40 | 39.30 | 35.80 | 57.50 | 85.10 | 75.20 | 40.40 | 45.10 | 36.60 | 34.20 | 52.80 |
|  | MRN(ICCV2023) [52] | 91.10 | **88.60** | 77.20 | 73.70 | 69.80 | 69.80 | 78.40 | 85.10 | **85.10** | 73.20 | 68.30 | 65.30 | 65.50 | 73.70 |
|  | HAMMER(Ours) | 91.12 | 87.97 | **79.26** | **76.56** | **76.16** | **74.78** | **80.98** | 85.09 | 84.04 | **75.31** | **71.28** | **71.34** | **70.31** | **76.23** |
| SVTR [7] | UpperBound | - | - | - | - | - | - | 90.10 | - | - | - | - | - | - | 83.20 |
|  | Baseline [52] | 90.60 | 32.50 | 40.50 | 30.80 | 24.50 | 19.90 | 39.80 | 84.80 | 31.30 | 37.00 | 29.20 | 22.60 | 19.10 | 37.30 |
|  | Baseline(Ours) | 91.12 | 51.52 | 29.26 | 22.15 | 18.26 | 14.38 | 37.78 | 85.09 | 51.68 | 28.33 | 21.11 | 16.42 | 13.72 | 36.06 |
|  | LwF(TPAMI2017) [18] | 90.60 | 28.00 | 38.40 | 29.90 | 24.10 | 18.30 | 38.20 | 84.80 | 27.00 | 34.60 | 28.40 | 22.30 | 17.00 | 35.70 |
|  | EWC(PNAS2017) [15] | 90.60 | 33.00 | 41.20 | 31.10 | 24.60 | 20.00 | 40.10 | 84.80 | 31.30 | 37.70 | 29.50 | 22.60 | 19.00 | 37.50 |
|  | WA(CVPR2020) [48] | 90.60 | 28.00 | 37.90 | 30.40 | 24.80 | 19.80 | 38.60 | 84.80 | 27.70 | 34.60 | 28.30 | 22.60 | 18.60 | 35.90 |
|  | DER(CVPR2021) [41] | 90.60 | 74.50 | 55.70 | 55.00 | 49.50 | 45.70 | 61.80 | 84.80 | 71.60 | 52.90 | 52.20 | 46.60 | 43.60 | 58.60 |
|  | MRN(ICCV2023) [52] | 90.60 | 86.40 | 73.90 | 65.60 | 63.40 | 58.10 | 73.00 | 84.80 | 83.70 | 69.40 | 64.40 | 57.80 | 53.10 | 68.90 |
|  | HAMMER(Ours) | 91.12 | **89.13** | **77.45** | **71.32** | **70.54** | **69.54** | **78.18** | 85.09 | **86.01** | **72.96** | **69.89** | **66.57** | **65.90** | **74.40** |
| TRBA [3] | UpperBound | - | - | - | - | - | - | 94.90 | - | - | - | - | - | - | 90.50 |
|  | Baseline [52] | 91.30 | 49.60 | 47.30 | 36.10 | 28.60 | 24.00 | 46.10 | 85.40 | 49.40 | 44.40 | 34.80 | 27.40 | 23.10 | 44.00 |
|  | Baseline(Ours) | 90.93 | 33.52 | 32.66 | 28.45 | 22.48 | 14.45 | 37.08 | 85.09 | 33.13 | 32.40 | 27.10 | 19.13 | 12.84 | 34.95 |
|  | LwF(TPAMI2017) [18] | 91.30 | 55.70 | 38.80 | 28.70 | 22.60 | 18.70 | 42.60 | 85.40 | 54.20 | 35.00 | 27.20 | 20.50 | 17.00 | 39.90 |
|  | EWC(PNAS2017)[15] | 91.30 | 50.40 | 43.60 | 33.10 | 25.60 | 21.90 | 44.30 | 85.40 | 49.40 | 40.60 | 31.70 | 24.80 | 20.60 | 42.10 |
|  | WA(CVPR2020) [48] | 91.30 | 45.40 | 41.80 | 30.70 | 23.50 | 19.60 | 42.10 | 85.40 | 44.00 | 37.90 | 29.20 | 21.60 | 18.10 | 39.40 |
|  | DER(CVPR2021) [41] | 91.30 | 60.10 | 53.00 | 38.80 | 31.40 | 28.60 | 50.50 | 85.40 | 60.70 | 50.30 | 37.20 | 30.30 | 28.10 | 48.70 |
|  | MRN(ICCV2023) [52] | 91.30 | 87.90 | 75.80 | 72.20 | 71.50 | 68.70 | 77.90 | 85.40 | 84.50 | 73.20 | 67.80 | 66.70 | 64.80 | 73.70 |
|  | HAMMER(Ours) | 90.93 | **89.02** | **80.62** | **78.92** | **78.37** | **77.75** | **82.60** | 85.09 | **86.34** | **76.56** | **74.45** | **73.65** | **73.10** | **78.20** |

**Table 3: Evaluation results of different components.**

| Index | Word-level | | Char-level | | MLT17 | | MLT19 | |
|---|---|---|---|---|---|---|---|---|
|  | New | Shared | New | Shared | Task6 | AVG | Task6 | AVG |
| 1 | ✗ | ✗ | ✗ | ✗ | 14.45 | 37.08 | 12.84 | 34.95 |
| 2 | ✓ | ✗ | ✗ | ✗ | 73.69 | 80.48 | 69.08 | 75.87 |
| 3 | ✓ | ✓ | ✗ | ✗ | 73.42 | 80.97 | 69.67 | 76.37 |
| 4 | ✗ | ✗ | ✓ | ✗ | 76.33 | 81.94 | 70.66 | 77.00 |
| 5 | ✗ | ✗ | ✓ | ✓ | 76.68 | 82.04 | 70.27 | 77.23 |
| 6 | ✓ | ✗ | ✓ | ✗ | 75.45 | 82.00 | 71.61 | 77.41 |
| 7 | ✓ | ✓ | ✓ | ✓ | 77.75 | 82.60 | 73.10 | 78.20 |

using standard cross-entropy, i.e., without considering incremental sharing situations. *Shared* indicates optimizing shared words and characters using multi-label loss. The baseline model (index-1) yields a relatively low average performance of 37.08% on MLT17 and 34.95% on MLT19. Comparing index-2, 3 and index-4, 5, we can see that predicting the belonging language scores only at the word level and only at the character level can improve the average performance, demonstrating the necessity of analyzing characters in text

recognition tasks. Analyzing shared words (index-3) and shared characters (index-5) can facilitate mining the dependencies between the old and the new language, thus minimizing the catastrophic forgetting of the old language, as observed when comparing index-2 with index-3 and index-4 with index-5. Ultimately, the HAMMER (index-7) performance is further improved when all the proposed sub-modules are optimized.

*4.3.2 Effect of Voting Ways.* In both the unified and specific guided prediction modules, a soft voting mechanism is employed to guide the final prediction. Soft voting involves using language scores as weights for individual text recognizers, which are then weighted and summed to obtain the final output. We also test the impact of hard voting on model performance, where the class with the largest probability of the DomainMLP output is taken as an indicator to select the corresponding specific recognizer. From Table 4, it can be observed that on MLT17 and MLT19, the results using soft voting at the word and char levels are much better than those using hard voting. Hard voting assumes that a word or a character belongs to a specific language, which contradicts multi-label optimization, resulting in poor performance. In contrast, the soft voting mechanism is consistent with the idea of multi-label optimization. It allows

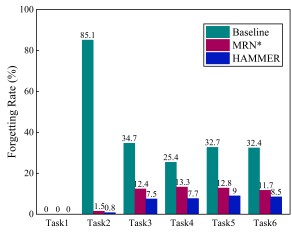
(a) Forgetting rate on MLT17

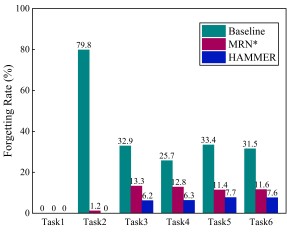
(b) Forgetting rate on MLT19

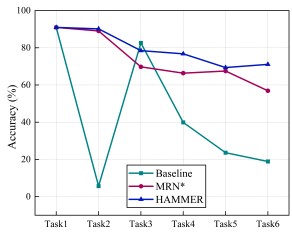
(c) Task 1 results on MLT17

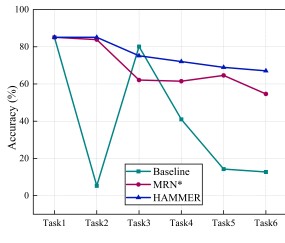
(d) Task 1 results on MLT19

**Figure 5: (a)(b) Forgetting rates of all tasks on MLT17 and MLT19. (c)(d) Results of initial task (task 1) on MLT17 and MLT19.**

**Table 4: Evaluation results for different voting ways.**

| Word-level | Char-level | MLT17 | | MLT19 | |
|---|---|---|---|---|---|
| | | Task6 | AVG | Task6 | AVG |
| Hard | Hard | 60.17 | 67.87 | 57.51 | 64.51 |
| | Soft | 73.34 | 81.01 | 69.21 | 76.44 |
| Soft | Hard | 72.90 | 80.55 | 69.14 | 76.00 |
| | Soft | 77.75 | 82.60 | 73.10 | 78.20 |

**Table 5: Evaluation results for different inference ways.**

| Inference | MLT17 | | MLT19 | |
|---|---|---|---|---|
| | Task6 | AVG | Task6 | AVG |
| Unified | 77.75 | 82.60 | 73.10 | 78.20 |
| Specific | 60.77 | 74.32 | 58.29 | 70.97 |
| Weighted | 74.67 | 81.95 | 70.68 | 77.77 |

for exploring dependencies between the old and the new language, mitigating catastrophic forgetting the most.

*4.3.3 Effect of Inference Output.* During training, the unified guided prediction module and the specific guided prediction module make predictions. In the inference process, it is crucial to determine which prediction should be chosen as the final output. We test three output strategies: the output guided by the word scores *Unified*, the output guided by the character scores *Specific*, and the output weighted by both the word and character scores *Weighted*. As shown in Table 5, the output guided by the word scores consistently yields superior results in the inference process. Therefore, we opt to utilize predictions guided by word scores as the final output.

*4.3.4 Effect of Incremental Order.* In addition to the default incremental order *Order-1*, we design two alternative incremental orders based on the number of training samples. *Order-2* and *Order-3* arrange the languages in the order of training samples from more to less and from less to more, respectively. The experimental results in Table 6 demonstrate that the choice of incremental order significantly influences the recognition performance. This observation underscores the importance of carefully selecting the incremental order, considering factors such as shared words and shared characters to optimize the overall performance.

**Table 6: Evaluation results for different orders. *O1* means Order-1, similar to *O2* and *O3*.**

| | Task1 | Task2 | Task3 | Task4 | Task5 | Task6 | Results |
|---|---|---|---|---|---|---|---|
| O1 | Chinese | Latin | Japanese | Korean | Arabic | Bangla | AVG |
| MLT17 | 90.93 | 89.02 | 80.62 | 78.92 | 78.37 | 77.75 | 82.60 |
| MLT19 | 85.09 | 86.34 | 76.56 | 74.45 | 73.65 | 73.10 | 78.20 |
| O2 | Latin | Korean | Japanese | Arabic | Bangla | Chinese | AVG |
| MLT17 | 94.26 | 90.25 | 84.76 | 84.84 | 81.56 | 74.33 | 85.00 |
| MLT19 | 92.83 | 86.58 | 79.37 | 80.21 | 77.03 | 70.55 | 81.09 |
| O3 | Chinese | Bangla | Arabic | Japanese | Korean | Latin | AVG |
| MLT17 | 90.93 | 87.38 | 85.08 | 76.50 | 73.05 | 74.84 | 81.27 |
| MLT19 | 85.09 | 81.03 | 80.39 | 73.00 | 68.03 | 70.66 | 76.37 |

## 4.4 Algorithm Analysis

*4.4.1 Analysis of Forgetting Rate.* The forgetting rate is a crucial performance measure for incremental learning methods. A lower forgetting rate indicates better retention of old knowledge by the model. We compute the forgetting rates for all MRN tasks based on the official code, denoted as MRN*. As illustrated in Fig. 5(a) and 5(b), our HAMMER exhibits a significantly lower forgetting rate than *Baseline* and MRN*. Furthermore, we statistically analyze the recognition accuracy for the initial task (task 1) during the incremental process, as shown in Fig. 5(c) and 5(d). The performance of *Baseline* on task 1 drops significantly when learning task 2, primarily due to the minimal shared knowledge between Latin (task 2) and Chinese (task 1), resulting in the failure to recognize Chinese. In contrast, our HAMMER exhibits the slowest degradation on the initial task, which can be attributed to the exploration of shared knowledge, enabling the mining of old knowledge and thus mitigating catastrophic forgetting.

## 5 CONCLUSION

We innovatively identify incremental sharing, characterized by shared words and characters among incremental languages. Building upon this observation, we propose HAMMER, a replay-based hierarchical multi-label learning framework for multilingual text recognition. A knowledge analysis module is designed to determine the shared knowledge and provide supervision. Additionally, we introduce a hierarchical language evaluation mechanism to guide specific recognizers to prediction sequences. Extensive experiments demonstrate the superiority of our HAMMER.

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
