# OpenReview forum: "Hierarchical Multi-label Learning for Incremental Multilingual Text Recognition"
_acmmm.org/ACMMM/2024/Conference — MM2024 Poster_

### Official Review · Reviewer_TBQP · 2024-05-08

**Rating:** 3
**Confidence:** 2

**Summary:**

This paper addresses the challenge of Multilingual Text Recognition (MLTR), particularly focusing on incremental learning scenarios where systems often grapple with preserving acquired knowledge when acquiring new languages. Introducing the concept of the "incremental sharing problem," the authors highlight the oversight of shared words and characters across languages in traditional incremental learning methodologies. To tackle this issue, they propose HAMMER, a Hierarchical Multi-label learning framework for Multilingual Text Recognition. HAMMER incorporates an online knowledge analysis system to identify and leverage shared language components, thereby enhancing the retention of old knowledge while assimilating new. By further implementing a hierarchical language evaluation strategy that operates at word and character levels, the framework promotes balanced learning and reduces catastrophic forgetting. Empirical validations on MLT17 and MLT19 datasets affirm the efficacy of HAMMER, establishing it as a new state-of-the-art solution in the field.

**Strengths:**

1. Novelty in Addressing Incremental Sharing: The paper identifies and defines a new problem domain within incremental learning specific to MLTR, contributing a fresh perspective to the field.
2. Hierarchical Multi-Label Learning Framework: HAMMER's design, integrating multi-level recognition and shared knowledge exploitation, is innovative and comprehensive, enhancing the model's adaptability and robustness.
3. Practical Impact: The proposed solution directly tackles a real-world challenge in global communication, making MLTR more efficient and adaptive to evolving multilingual contexts.
4. Empirical Validation: Strong experimental results on main benchmark datasets validate HAMMER's superiority, underlining its practical significance and advancing the SOTA.

**Limitations:**

1. Scalability Concerns: The effectiveness of HAMMER with an extensive number of languages or in scenarios with rapidly growing shared vocabulary sizes is not thoroughly explored.
2. Generalizability to Low-Resource Languages: The paper does not explicitly discuss how HAMMER would perform with low-resource languages that may lack sufficient shared words or characters with previously learned languages.
3. Complexity of Implementation: The hierarchical and multi-label learning components introduce added complexity, which may impact computational requirements and potentially hinder swift adoption or implementation by researchers or developers with limited resources.
4. Assessment of Shared Knowledge Identification: While the knowledge analysis module is a key feature, the paper does not delve deeply into its precision or robustness, leaving room for further scrutiny of its effectiveness in varied linguistic contexts.

**Suitability:**

2

---

### Official Review · Reviewer_LxZ4 · 2024-05-15

**Rating:** 4
**Confidence:** 3

**Summary:**

This work conducts research on scene text recognition tasks in multi-language text recognition. The authors observe that effective learning of words and characters shared between languages has been neglected in previous work. Inspired by this observation, this work proposes a hierarchical multi-label learning framework for multilingual text recognition, called HAMMER. Online knowledge analysis aims to identify shared knowledge and provide corresponding multi-label language supervision. Specifically, only words and characters that appear simultaneously in multiple languages are considered shared knowledge. Furthermore, to further capture language dependence, this work introduces a hierarchical language assessment mechanism to predict word and character level language scores. These scores, under the supervision of knowledge analysis, guide specific recognizers to effectively exploit old and new language knowledge, thereby mitigating catastrophic forgetting due to imbalanced rehearsal sets.

**Strengths:**

This work starts from an observed phenomenon, gets inspiration, and then designs a model.
From the perspective of model design, this job is a good one, with its own ingenuity and uniqueness.
From the perspective of experimental implementation, this work has produced a large number of experiments, and the experimental design is reasonable and comprehensive.

**Limitations:**

The shortcomings of this work include:
1. In the era of deep learning, the ambiguity of shared words and characters will be alleviated due to contextual information. Is it appropriate to treat it as a multi-label learning task?
2. The statistics of words and characters in the dataset are given in Table 1. Except for Chinese, the proportion of tasks 2, 3, and 4 in the training set and test set is greater than that of tasks 5 and 6, but this pattern does not appear in the experimental results.
3. The experimental comparison models are relatively old, and two of the five comparison models are 16 years old.

**Suitability:**

3

---

### Official Review · Reviewer_YFMa · 2024-05-24

**Rating:** 5
**Confidence:** 3

**Summary:**

The paper focuses on the problem of Multilingual Text Recognition (MLTR), which is essential for facilitating cultural communication. Existing methods often struggle with retaining previous knowledge when learning new languages, which leads to the problem of catastrophic forgetting. HAMMER aims to mitigate this by using shared knowledge across languages, particularly through words and characters that appear simultaneously in multiple languages.

Specifically, HAMMER introduces a hierarchical approach that leverages shared words and characters across multiple languages. This shared knowledge is used to predict language scores at both word and character levels, which helps in retaining previously learned knowledge and acquiring new language knowledge.

The paper presents extensive experiments conducted on benchmark datasets ML17 and ML19, showing that HAMMER performs better than existing state-of-the-art approaches.

**Strengths:**

Novel Framework: HAMMER's hierarchical approach is innovative and addresses a significant challenge in MLTR by leveraging shared knowledge across languages.

Comprehensive Analysis: The paper provides a thorough analysis of shared knowledge and how it can be effectively used to predict language scores.

Promising Results: The experimental results on benchmark datasets demonstrate the robustness and effectiveness of the proposed method.

**Limitations:**

Though the idea of knowledge sharing sounds reasonable, the reviewer feels unclear of why it can work, especially for languages that have completely different styles of characters, i.e., English and Chinese. It will be helpful if the authors can share more insights or illustrations of how knowledge is transferred.

The studies in Table 6 are interesting; the order of learning matters for the final performance. Such an observation might partially prove the difficulty of transferring knowledge between languages of different characters. The reviewer suggests adding more orders and analyses.

Overall, I think this paper is solid and can bring value to the community. But the presentation should be improved, for example:

In Equations (4) and (5) on line 427, the authors use superscripts w and c to represent word-level and character-level, while the definition of them is introduced on line 509. This is confusing.

The authors are encouraged to put summarized sentences in table/figure captions, rather than simply drawing/listing the results.

**Suitability:**

3

---

### Meta-Review · Area_Chair_EzVC · 2024-07-01

**Recommendation:** Accept (Poster)
**Confidence:** 4

**Metareview:**

This paper proposes HAMMER, a novel Hierarchical Multi-label learning framework designed to address Multilingual Text Recognition (MLTR) by leveraging shared words and characters across languages to mitigate catastrophic forgetting. HAMMER's innovative approach, combined with its hierarchical evaluation strategy, shows promising results on benchmark datasets ML17 and ML19, outperforming state-of-the-art methods. While the framework's novelty and comprehensive analysis are commendable, the paper would benefit from addressing several reviewer concerns. Specifically, clarifying the mechanism of knowledge transfer between languages with different character styles, providing more insights into the effectiveness of shared knowledge, and enhancing the clarity and organization of the presentation are necessary. Additional experiments with various learning orders and comparisons with recent models would further validate the approach. Addressing these points in the camera-ready version will strengthen the paper's contributions and impact on the MLTR field.